# Recent Advances in Pre-Clinical Development of Adiponectin Receptor Agonist Therapies for Duchenne Muscular Dystrophy

**DOI:** 10.3390/biomedicines12071407

**Published:** 2024-06-25

**Authors:** Shivam Gandhi, Gary Sweeney, Christopher G. R. Perry

**Affiliations:** 1School of Kinesiology and Health Science, Muscle Health Research Centre, York University, Toronto, ON M3J 1P3, Canada; shivamg@yorku.ca; 2Department of Biology and Muscle Health Research Centre, York University, Toronto, ON M3J 1P3, Canada; gsweeney@yorku.ca

**Keywords:** Duchenne muscular dystrophy, cardiomyopathy, inflammation, skeletal muscle, adiponectin

## Abstract

Duchenne muscular dystrophy (DMD) is caused by genetic mutations in the cytoskeletal-sarcolemmal anchor protein dystrophin. Repeated cycles of sarcolemmal tearing and repair lead to a variety of secondary cellular and physiological stressors that are thought to contribute to weakness, atrophy, and fibrosis. Collectively, these stressors can contribute to a pro-inflammatory milieu in locomotor, cardiac, and respiratory muscles. Given the many unwanted side effects that accompany current anti-inflammatory steroid-based approaches for treating DMD (e.g., glucocorticoids), there is a need to develop new therapies that address inflammation and other cellular dysfunctions. Adiponectin receptor (AdipoR) agonists, which stimulate AdipoR1 and R2 isoforms on various cell types, have emerged as therapeutic candidates for DMD due to their anti-inflammatory, anti-fibrotic, and pro-myogenic properties in pre-clinical human and rodent DMD models. Although these molecules represent a new direction for therapeutic intervention, the mechanisms through which they elicit their beneficial effects are not yet fully understood, and DMD-specific data is limited. The overarching goal of this review is to investigate how adiponectin signaling may ameliorate pathology associated with dystrophin deficiency through inflammatory-dependent and -independent mechanisms and to determine if current data supports their future progression to clinical trials.

## 1. Overview of Muscular Dystrophy 

### 1.1. Introduction

Muscular dystrophies (MDs) are hereditary or genetic (spontaneous/non-hereditary) diseases characterized by severe progressive deterioration of locomotor, respiratory, and/or cardiac muscles. While there are many types of muscular dystrophies, X-linked dystrophinopathy, defined by over 70 different mutations in the dystrophin gene, is the most common and occurs predominantly in males [1,2]. Such mutations can lead to a complete absence of dystrophin (Duchenne muscular dystrophy; DMD) or a truncated transcript (Becker muscular dystrophy; BMD) [3,4,5,6]. While DMD is generally more severe than BMD, both diseases cause muscle weakness, atrophy, and fibrosis, leading to reduced mobility, respiratory and cardiac dysfunction, as well as reduced lifespans. Dystrophin mutations often result in additional systemic dysfunctions, including cognitive impairment, digestive abnormalities, anxiety, depression, or obesity [7,8,9]. 

While there is no cure, ongoing pre-clinical research seeks to restore the normal genetic sequence through emerging gene editing technologies such as CRISPR or produce truncated dystrophin transcripts with exon skipping and microdystrophin gene therapies [10,11,12]. Exon skipping therapy was recently approved by the FDA in the USA for certain mutations, but the technology must be adapted for each of the numerous mutations. Their ability to partially improve muscle dysfunction underscores the importance of maintaining glucocorticoid therapy as a major standard of care, given that systemic inflammation is a major contributor to muscle dysfunction [13,14,15] for virtually all persons with DMD. While the effectiveness of glucocorticoids in slowing the decline of muscle function demonstrates the value of targeting secondary contributors to these diseases, the eventual decline in muscle function and side effects (i.e., attenuated growth, obesity, mood disorders, other) [16] underscores the importance of developing additional treatments that provide benefits for most patients regardless of the specific underlying mutation.

Exogenously administered agonists of adiponectin (ApN) and its downstream cell-surface receptors have emerged as an attractive candidate for pharmacological intervention given their anti-inflammatory properties in DMD [17,18,19,20,21,22] and ability to induce metabolic reprogramming in muscle [10,23,24,25,26,27,28,29]. Recently published pre-clinical data from in vivo rodent and in vitro human cell-based models suggests that AdipoR agonists may be beneficial for attenuating some of the secondary physiological stressors associated with DMD. In this regard, the purpose of this review is to provide an overview of the pathophysiology of DMD with a perspective of translating disease mechanisms to the development of ApN agonist therapies in order to stimulate discussion in this new area of research. 

### 1.2. Muscle Damage and Inflammation in DMD

Dystrophin is an indispensable component of cellular architecture, given that it is responsible for anchoring the actin cytoskeleton to the dystrophin-associated glycoprotein complex (DGC) at the cell membrane (sarcolemma) to maintain cellular stability [30]. Dystrophin-deficient models are characterized by loss of myofiber integrity, essentially rendering muscle fibers susceptible to contraction-induced damage [31]. In DMD, muscle fibers demonstrate chronic damage that arises from contraction–relaxation cycles. However, the regenerative capacity of muscle fibers eventually exhausts, leading to impaired homeostatic muscle repair and turnover [32,33,34]. This cycle leads to muscle fiber necrosis and fibrofatty replacement of necrotic tissue, as well as atrophy, as a last-resort mechanism to uphold the cytostructure [35]. 

Key to the pathology of DMD is the prolonged activation of the innate immune system in response to the chronic contraction-induced damage of muscle fibers [36]. The innate immune response is triggered when granulocytes, monocytes, monocyte-derived macrophages, and dendritic cells are triggered by damage-associated molecular patterns (DAMPs) that leak from damaged muscle fibers [37,38]. DAMPs trigger the recruitment of macrophages and neutrophils to sites of damage by binding to their pathogen recognition receptors (PRRs), which include toll-like receptors (TLR2/4/7) [33,38,39,40,41]. Interestingly, data have demonstrated that deleting TLR2 or administering a TLR7/9 antagonist in C57BL/10.*mdx* mice, which is a commonly utilized rodent model of DMD, reduces muscle inflammation and improves skeletal muscle function, thus supporting the notion that PRRs play a pivotal role in promoting muscle degeneration [42,43]. 

Following TLR activation, downstream inflammatory signaling is mediated by nuclear factor kappa B (NF-κB) [44], c-Jun NH_2_-terminal kinase (JNK) [45], and interferon regulatory factors (IRFs), which are activated by tumor necrosis factor-alpha (TNFα) [46], interleukin (IL) 6 (IL-6) [47], and the myeloid differentiation primary response 88 (MyD88)-dependent pathways [34,39]. NF-κB activation induces the expression of pro-inflammatory genes in the nucleus [37,39], including IL-6, which promotes inflammation. IL-6 also interferes with muscle satellite cell populations and impedes muscle regeneration [48,49]. 

The induction of pro-inflammatory signaling events occurs in M1 classically-activated macrophages [36,50]. Since DMD, by definition, is characterized by asynchronous cycles of muscle damage and repair, M1 macrophages must be continuously recruited to sites of damage to sustain an immune response. Consequently, a high concentration of pro-inflammatory cytokines such as TNFα, IL-6, and IL-1β perpetuates a chronic inflammatory state [51]. Although many different chemo-attractive molecules can stimulate the recruitment of immune cells to dystrophic muscles, C-C motif chemokine receptor type 2 (CCR2) has demonstrated a significant role in recruiting inflammatory cells to sites of injury in C57BL/10.*mdx* muscle [32,36]. During early phases of inflammation, elevated pro-inflammatory cytokine concentrations can lead to the production of inducible nitric oxide synthase (iNOS), which, alongside other cytoplasmic and mitochondrial oxidizing radicals [52,53], can significantly damage dystrophin-deficient skeletal muscle by increasing damage to surrounding tissues and causing aberrant cell lysis [50]. While M1 pro-inflammatory macrophages generally induce damage, M2 CD206-expressing alternatively-activated anti-inflammatory macrophages release anti-inflammatory cytokines like IL-10, IL-4, and insulin-like growth factor-1 (IGF-1), which downregulate iNOS production and promote muscle repair in dystrophin-deficient muscle [50]. Among the many responsibilities of M2 macrophages, they are vital for regulating skeletal muscle regeneration by ensuring the proliferation and maturation of muscle progenitor cells, which include satellite cells and collagen-secreting fibroblasts [54]. 

These steps culminate in two major mechanisms regulating muscle dysfunction. First, inflammation inhibits muscle satellite cells and regeneration [55]. Second, continual recruitment of M2 macrophages leads to increased release of transforming growth factor beta (TGFβ) that stimulates fibroblast activity and production of extracellular matrix (ECM) proteins, including excessive collagen, to create a form of ‘reactive fibrosis’ [56]. The balance between classically activated M1 populations and alternatively-activated M2 populations remains critical to consider when examining processes that maximize the reparative potential of muscle. 

Neutrophils remove cellular debris that accumulates in damaged regions [57]. Studies have shown that neutrophils are recruited to sites of injury at early stages of dystrophinopathy in C57BL/10.*mdx* mice and can be approximately 30% more numerous than macrophages in dystrophic muscle [50,58]. Despite the protective properties of neutrophils in healthy physiological systems, they can also impair regeneration in dystrophic muscle by stimulating the secretion of myeloperoxidase (MPO), which is predominantly involved in catalyzing the production of hypochlorous acid (HOCl)—a damaging and reactive oxidant—in the presence of hydrogen peroxide (H_2_O_2_) and chloride (Cl^-^) at sites of inflammation [36,59,60]. Data have shown that golden retriever muscular dystrophy (GRMD) muscle exhibits significantly higher levels of MPO compared to healthy WT muscles, thus suggesting that neutrophil-derived MPO might be contributing to muscle damage by inducing oxidative stress [59]. In addition to MPO release, proteomic analyses of C57BL/10.*mdx* muscle have also revealed elevated production of neutrophil elastase (NE) compared to healthy WT muscle [61]. NE can be particularly damaging in dystrophic muscle due to its propensity to impair myoblast survival and proliferation by promoting cell adhesion molecule (CAM) degradation [61]. 

Cytotoxic T-lymphocytes, or the CD4^+^ T cells, have been identified as additional mediators of the dystrophic immune response. These cells, which can be subdivided into regulatory T cells (Tregs) and conventional T helper (Th) cells, have been associated with reductions in muscle inflammation and damage in dystrophic muscle [36]. The differentiation of Tregs from naïve T cells is controlled by the transcription factor FoxP3, whose expression is induced by TGFβ [36]. The presence of Tregs has generally been cited as being beneficial for dystrophic muscle, given that they are immunosuppressive and express the anti-inflammatory cytokine, IL-10 [36]. Two separate models, one employing rapamycin-treated C57BL/10.*mdx* muscle to demonstrate elevated Tregs [62] and the other depleting Tregs via antibody depletion of CD25^+^ cells [63], implicated Tregs in reducing muscle fiber damage, managing serum creatine kinase (CK) concentrations, and also reducing muscle inflammation and interferon γ (IFNγ) expression [36,50]. Additionally, ablation of FoxP3-expressing Tregs exacerbated *mdx* muscle damage and led to elevated IFNγ expression and reductions in the expression of the M2 macrophage-specific marker, CD206 [50]. Collectively, these data suggest that although Tregs exist at an extraordinarily low frequency of occurrence in sites of damage, they still play a vital role in the transitionary stages of early C57BL/10.*mdx* pathology to later-onset regenerative stages [36]. 

Many of these inflammatory responses are attenuated by common glucocorticoids used as standard of care (e.g., prednisolone) for DMD. For example, a study conducted by [64] investigated the effects of prednisolone treatment in C57BL/10.*mdx* mice between 2 and 4 weeks of age on several immune-related markers, including pro-inflammatory macrophages (using an anti-F4/80 marker), CD4^+^ T cells, and CD8^+^ T cells in quadriceps and soleus muscle. The group determined that prednisolone treatment reduced F4/80-positive macrophages (57–59% reduction), CD4^+^ T cells (50–60% reduction), and CD8^+^ T cells (48–58% reduction) in both muscles [64], suggesting that glucocorticoids may play an essential role in modulating the DMD-induced immune response. The effects of glucocorticoids on macrophage markers and T cell activation have been well characterized in several other pathological models with robust inflammation as well [65,66]. Interestingly, work investigating the effects of glucocorticoid treatment on human peripheral lung inflammation in asthma identified significant elevations to neutrophils following treatment, accompanied by improvements in lung function [67], which may attest to the beneficial/pleotropic role of neutrophil recruitment at sites of injury and damage, such as in DMD. However, the side effects of glucocorticoids warrant consideration of alternative therapies. As mentioned previously, ApN is an attractive candidate for consideration in DMD due to its anti-inflammatory properties; however, the mechanisms through which these anti-inflammatory effects are elicited still require elucidation, as discussed in the next section. Understanding the role of ApN in various pathological models serves as a foundation for investigating how exogenous ApN administration might be efficacious in DMD for attenuating secondary physiological stressors. 

## 2. Physiological Role of Adiponectin in Non-Dystrophic Models

### 2.1. Cellular Properties and Structural Features

ApN and its receptors (which exist as structurally related AdipoR1 and AdipoR2 isoforms) have been implicated in a myriad of functions, ranging from regulating cellular metabolism to anti-inflammatory effects in both skeletal muscle and cardiac tissue [21,68,69]. In addition to AdipoR1 and AdipoR2, T-cadherin (T-Cad) is another cell surface molecule that has a significant affinity for high molecular weight oligomers of ApN [70]. Although T-Cad can bind to ApN, its lack of intracellular signaling domain impedes its consideration as an ApN signaling receptor [70]. 

ApN is a 30-kDa multimeric protein, abundantly secreted by mature adipocytes within white adipose tissue (WAT), that consists of a globular C-terminal domain and a collagen-like N-terminal domain [70,71,72]. Although the collagenous domain allows ApN to be secreted into the bloodstream as three oligomeric complexes, including a low molecular weight (LMW) trimer form (67 kDa), middle molecular weight (MMW) hexamer form (140 kDa), and high molecular weight (HMW) (300 kDa) form [73], the HMW oligomer in particular elicits insulin-sensitizing and cardioprotective properties [74]. While the trimer is formed by hydrophobic interactions in its globular heads and is stabilized by non-covalent interactions of the collagen-like domains, the hexamer and HMW forms of ApN require intermolecular disulfide bond formation between highly conserved cysteine residues [74,75]. This is particularly important because several post-translational modifications, including hydroxylation and glycosylation on conserved lysine residues, are vital for the assembly and secretion of HMW ApN [74]. 

While ApN is produced by WAT in its full-length (fAd) form, fAd can be proteolytically/enzymatically cleaved to the smaller globular (gAd) form by neutrophil elastase produced from monocytes or macrophages [76,77]. In addition to secretion by WAT, ApN can also be secreted by human and murine liver parenchymal cells [78], skeletal muscle myocytes [79], cardiac epithelial cells [80], endothelial cells [81], osteoblasts [71,82], and kidney tubular cells [83]. ApN can represent between 0.01 and 0.05% of total plasma protein (~2–20 μg/mL range) in humans, thus attesting to its abundance in circulation [70,84,85,86]. Despite being a stable protein in plasma, ApN has a relatively short half-life in circulation of only ~45–75 min in humans and is cleared primarily by the liver [87]. 

ApN is an attractive therapeutic candidate for a host of pathologies [88], given that its circulating levels have been negatively correlated with cardiovascular disease (CVD), cancer, and metabolic syndrome, while high circulating concentrations have been correlated with healthy physiological systems. In this regard, extensive research has demonstrated that ApN is an important regulator of carbohydrate and fat metabolism in multiple tissues, as described below.

### 2.2. Multi-Organ and Inflammatory Regulation of ApN

Through interactions with hepatic AdipoR1 and AdipoR2 receptors, ApN exerts regulatory control over glucose uptake and fat metabolism by reducing hepatic gluconeogenesis [89,90], glycogenolysis [91], and lipogenesis [24,92] while enhancing hepatic fatty acid oxidation [23,93]. 

AdipoR1 is the predominant adiponectin receptor isoform found in skeletal muscle [24]. ApN regulates skeletal muscle glucose and fat metabolism and insulin sensitivity in part through AMPK-p38-MAPK signaling and PPAR-α induction [24,94,95]. Since ApN is able to activate AMPK by interacting with AdipoR1 and its downstream adaptor protein < adaptor protein, phosphotyrosine interacting with PH domain and leucine zipper 1 (APPL1) > [90], it is able to stimulate fatty acid oxidation and glucose entry into muscle cells [23,93]. 

The cardioprotective properties of ApN can be partly attributed to its effects on cardiac metabolism, apoptosis, autophagy, and hypertrophy [88,96]. The pleotropic roles of ApN as a modulator of cardiac pathology have been reviewed extensively elsewhere [96,97,98]. Briefly, through AdipoR1–APPL1 interactions, ApN signals AMPKα2, which regulates fatty acid β-oxidation by stimulating acetyl-CoA carboxylase (ACC) phosphorylation, thus enhancing fatty acid oxidation in the heart [99]. Interestingly, one study demonstrated that ApN possesses the ability to influence intramyocellular Ca^2+^ transients in cardiomyocytes from adult mice subjected to myocardial ischemia/reperfusion injury by reversing the decrease in Ca^2+^/calmodulin-dependent protein kinase-phospholamban (CaMKII-PLB) phosphorylation and sarcoendoplasmic reticulum calcium ATPase (SERCA2) activity [100]. This influence on Ca^2+^ transients in cardiomyocytes may be cardioprotective in theory, given that intramyocellular Ca^2+^ regulates contractile force output. The relationship between ApN and intramyocellular Ca^2+^ (at least in non-DMD models of cardiac stress) is particularly curious, given that DMD is characterized by disrupted sarcolemmal ion channel homeostasis and resultant Ca^2+^ overload. Future research should seek to investigate this relationship to determine if the pleiotropic roles of ApN interact to create a net effect on muscle health in *mdx*/DMD that is cardioprotective or further exacerbates the DMD-induced pathology. It should also be noted that a similar effect of ApN to induce intramyocellular Ca^2+^ influx to skeletal muscle for the activation of CaMKII, AMPK, and mitochondrial biogenesis has been observed, albeit once again in non-dystrophic muscle [90,100,101]. 

Although the mechanisms by which ApN contributes to improvements in cardiac pathology are multifactorial and generally implicate AMPK and Cyclooxygenase-2 (COX2) signaling [102], limited data are available on the intersection between ApN, cardiac function, and DMD. This gap in knowledge represents an avenue for future research, where additional work is encouraged to explore the role of ApN on cardiac electrophysiology, alterations to echocardiographic parameters, fibrosis, Ca^2+^ transients, and inflammation as common diagnostic indicators of DMD-induced cardiomyopathy. 

ApN has long been recognized for its propensity to shift macrophage polarization toward an M2 anti-inflammatory phenotype (Figure 1), at least in murine peritoneal cavity and adipose tissue [103]. Of note, ApN induces the production of the anti-inflammatory cytokine IL-10 while also suppressing the growth and proliferation of bone marrow-derived macrophage progenitors in human leukocytes [70,104,105]. While macrophages from ApN-null mice demonstrate an M1 pro-inflammatory phenotype by releasing higher levels of tumor necrosis factor (TNF-α), monocyte chemoattractant protein (MCP-1), and IL-6 compared to WT mice, the elevations to these cytokines is reversed by exogenous recombinant ApN administration [103] (Figure 1). Accordingly, while ApN is able to suppress differentiation and classical activation of M1-like pro-inflammatory macrophages, it is also able to promote M2-like ‘alternatively-activated’ anti-inflammatory macrophage proliferation and expression of cytokines like IL-10 (Figure 1), and the M2 macrophage markers Arg-1and Mgl-1 [103,104,106,107,108] (Figure 2). To date, little is known about the role of ApN in regulating T cell and neutrophil function. Although previous studies have demonstrated that AdipoR1 receptors are expressed in murine Tregs [109], the mechanism through which this relationship confers anti-inflammatory effects requires further insight. 

## 3. Synthetic Adiponectin Receptor Agonists

The discovery and design of AdipoR agonists that activate downstream signaling cascades have attracted great effort, given that it is difficult and expensive to produce biologically active recombinant ApN with an optimized dosage and route of administration for pre-clinical or clinical use [88] due to its large multimeric size and requirement for extensive post-translational modifications [110]. Several molecules that activate AdipoR have been developed and explored in a variety of conditions including pre-clinical models of DMD. 

### 3.1. AdipoRon

AdipoRon is a synthetic small-peptide AdipoR agonist that acts via both AdipoR1 and AdipoR1 to exert ApN-like effects [111,112]. It is the most studied AdipoR agonist available. Given that AdipoRon was the first orally active endogenous AdipoR agonist, many studies have been conducted to test its efficacy across different pathologies. Rodent models utilizing AdipoRon have determined that it has the ability to ameliorate insulin resistance, diabetes, and inflammation, in addition to its antiproliferative properties in various cancer models [113,114]. Subsequent studies demonstrated limited potency and specificity, which prompted the development of other AdipoR agonists [111].

### 3.2. ALY688

The 10 amino acid-long peptidomimetic ADP355 was developed following the identification of the critical receptor binding domain of ApN. This compound demonstrated high specificity for both receptor isoforms, was modified to be more resistant to proteolytic degradation, and demonstrated greater potency than ApN [112,115,116]. Now called ALY688, this small peptide was shown to lower inflammation in inflammatory disorders such as dry eye disease and reduce fibrosis in the liver [94,117,118]. Recent work using ALY688 has demonstrated its efficacy at increasing basal glucose uptake and enhancing insulin-stimulated glucose uptake in skeletal muscle cells while also improving glucose handling when administered to mice on a high-fat, high-sucrose diet [94]. Additionally, the same group also determined that daily subcutaneous ALY688 administration to 10–12-week-old C57BL/6 mice subjected to pressure overload attenuated cardiac hypertrophy, cardiac remodeling, fibrosis, and several cytokines consistent with inflammation, including IL-6, TLR-4, and IL-1β [119]. 

## 4. Pre-Clinical Development of Adiponectin-Receptor Agonists for DMD

Boys with DMD have lower serum ApN concentrations [120], as do male *mdx* mice [121]. As such, several investigations have explored the potential of AdipoR signaling to prevent inflammation, metabolic dysfunction, and fibrosis in pre-clinical models of DMD. 

Using C57BL/10.*mdx* mice, overexpression of ApN improved whole-body measures of muscle function, including grip strength, wire test, and treadmill activity, while reducing muscle damage and markers of inflammation [17]. AdipoRon treatment in this same model of DMD reduced inflammatory markers and muscle damage in skeletal muscle, stimulated markers of regeneration, lowered general marks of oxidative stress, and improved similar whole-body tests of muscle function [18]. This foundational work provides novel evidence to suggest that ApN signaling can beneficially influence whole-body indices of dystrophin deficiency-induced damage. Specifically, the work demonstrates a relationship between both genetically and pharmacologically elevated ApN in C57BL/10.*mdx* mice and attenuations of the dystrophic phenotype. These findings are important because they position ApN as a prospective therapeutic target and potential biomarker to aid in addressing DMD.

Two recent studies explored the potential for ALY688 to modify the disease process in mouse models of DMD. In one study, daily subcutaneous administration of ALY688 for 2 months beginning at 4 weeks of age in C57BL/10.*mdx* mice improved treadmill activity, wire test, and grip strength while lowering tibialis anterior necrosis and fibrosis [21]. Reductions in IL-1β and TNFα, the M1-type macrophage marker CD68^+^, as well as the lipid peroxidation product 4-HNE (4-Hydroxynonenal) were reduced in the limb muscle gastrocnemius. ALY688 treatment increased the expression of muscle differentiation and maturation factors and enhanced the myogenic program, leading to partial increases in revertant dystrophin-expressing fibers in the quadriceps. mRNA levels of *Mrf4*, a marker of late muscle differentiation, was halved in C57BL/10.*mdx* mice and partially restored with ALY688 as were other regeneration markers (Figure 2). ALY688 also lowered fibrosis in quadriceps and the fibrotic regulators TGFβ and p-SMAD2 while activating adenosine monophosphate kinase (AMPK), which has been shown to mediate AdipoR-mediated reductions in inflammation and fibrosis and stimulate both glucose and fatty acid oxidation [19,24,101] (Figure 2). Incubation of myotubes derived from DMD patients with ALY688 lowered IL-1β and TNFα and increased the expression of utrophin, a dystrophin homolog—the latter also being increased in the treated C57BL/10.*mdx* tibialis anterior muscle. As discussed by the authors [21], the collective results suggest a possible role for ALY688 in lowering inflammation and fibrosis through AMPK activation, given that this mechanism was previously demonstrated in C57BL/10.*mdx* mice [122]. 

In a second study, ALY688 was injected daily into D2.*mdx* mice—a more severe model of DMD than the C57BL10.*mdx* model—beginning at day 7 of age up to day 28 of age in order to determine the early effects of AdipoR agonism on muscle remodeling [22]. In the diaphragm, treatment reduced fibrosis in relation to lower IL-6 mRNA but increased IL-6 and TGFβ protein contents. ALY688 lowered mitochondrial complex I-stimulated H_2_O_2_ emission (a form of reactive oxygen species; ROS) without restoring pyruvate oxidation (a marker of glucose oxidation) that was shown to be lower in untreated D2.*mdx* mice assessed with high-resolution respirometry in vitro. Treatment lowered diaphragm force production assessed in vitro while quadriceps were not affected and remained lower compared to WT mice. Serum CK—a marker of sarcolemmal damage—was decreased by high doses of ALY688 and increased with low doses, while grip strength, cage hang time, and voluntary wheel running were unaffected by treatment. No changes in AMPK phosphorylation were observed but it was noted that future studies could consider whether the signaling effects are rapid and transient and would be captured by assessing tissues shortly after the last drug injection. The early prevention of diaphragm fibrosis and markers of muscle damage at high doses of treatment in these 4-week-old mice suggests future studies could examine longer-term treatment into adulthood when complex muscle development during adolescence is complete. 

A separate study (using a parallel study design) assessed the effects of daily ALY688 treatment on indices of recognition memory in D2.*mdx* mice [123]. Interestingly, ALY688 prevented impairments in recognition memory in D2.*mdx* mice, indicated by a lower discrimination index in the novel objection recognition test. Furthermore, decreased hippocampal mitochondrial pyruvate-supported respiration was also rescued by ALY688. Lastly, ALY688 completely prevented increases in protein contents of the neurofibrillary tangles marker total Tau, as well as protein contents of the upstream marker of tangles and autophagy, total Raptor. Together, the results indicate that ApN agonism improves recognition memory in D2.*mdx* mice through mechanisms that are not yet fully elucidated. 

Collectively, ALY688 demonstrates contrasting effects on inflammatory cascades that may be dependent on the age of assessments, the mouse model employed, the duration of treatment, or the muscles selected for analyses. While no studies of ApN/AdipoR agonists have been performed in humans, some work has assessed the effects of ApN on human DMD-derived myotubes separate from the lower IL-1β and TNFα responses to ALY688 treatments discussed above [21]. Specifically, ApN incubations reduced several pro-inflammatory cytokines, including TNFα, IL-17A, and CCL28 [19] (Figure 2), while also decreasing the NLRP3 inflammasome [20]. ApN treatment in human DMD-derived myotubes increases IL-6 [19], similar to the findings of ALY688 treatment in young D2.*mdx* mice discussed above [22], but in contrast to the effects seen with longer treatments and later ages in C57BL10.*mdx* mice [21]. The repression of NLRP3 is notable given that *mdx*/NLRP3-KO cross-bred mice reverse the increases in caspase-1 activity and contents of the pro-inflammatory cytokines IL-1β, IL-18, and TNFα seen in C57BL/10.*mdx* controls [20]. 

## 5. Conclusions and Future Directions

Pre-clinical studies have demonstrated improved skeletal muscle quality following overexpression of ApN or treatment with AdipoRon or ALY688 in *mdx* mouse models of DMD. The precise mechanisms are not fully elucidated, although reduced inflammation and metabolic reprogramming have been identified in some but not all of these approaches. Careful consideration of specific pre-clinical models, stage of disease, treatment duration, and muscle heterogeneity should be considered when constructing experimental designs with direct assessments of muscle force-generating capacities complemented with whole-body functional assessments. The divergent responses of cytokines between studies investigating ALY688 in *mdx* mouse models of DMD, despite a consistent response of lower fibrosis, may be consistent with prior reports that show specific cytokines can be both pro- or anti-inflammatory depending on specific contexts [124], which underscores the complexities of cytokines in remodeling dystrophic muscle during disease progression. Furthermore, resolving the apparent complexities in the inflammatory and metabolic responses to these approaches will give new insight into the fundamental mechanisms by which adiponectin receptors modify disease development in pre-clinical models, in addition to guiding translational efforts toward clinical trials. In this regard, the considerable expense of synthesizing ApN due to its large size and requirement for extensive post-translational modification [116] and the limited potency and specificity of AdipoRon [113] limits their potential for further clinical development. Peptidomimetics such as ALY688 have potential for clinical development and, therefore, warrant further consideration for pre-clinical research throughout the disease process in DMD. 

## Figures and Tables

**Figure 1 biomedicines-12-01407-f001:**
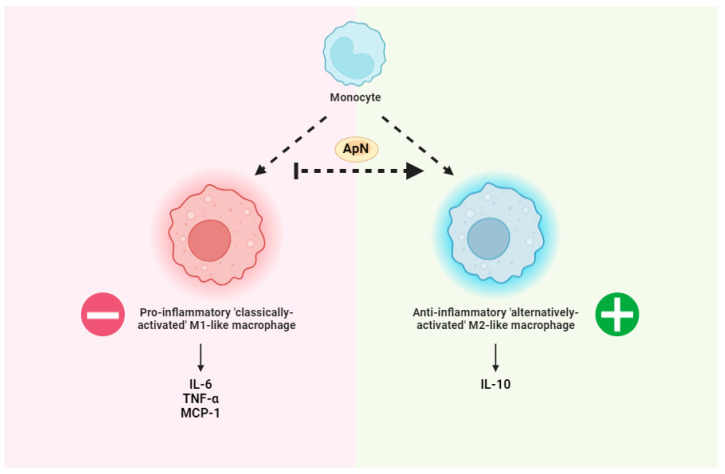
General overview of adiponectin (ApN)-mediated reductions in inflammation. ApN promotes the differentiation of monocytes to anti-inflammatory ‘alternatively activated’ M2-likemacrophages. This event is associated with lower pro-inflammatory cytokines including IL-6, TNF-α, and MCP-1 as well as increases in the anti-inflammatory cytokine IL-10 [103], as examples. Made using www.biorender.com (accessed on 1 June 2024).

**Figure 2 biomedicines-12-01407-f002:**
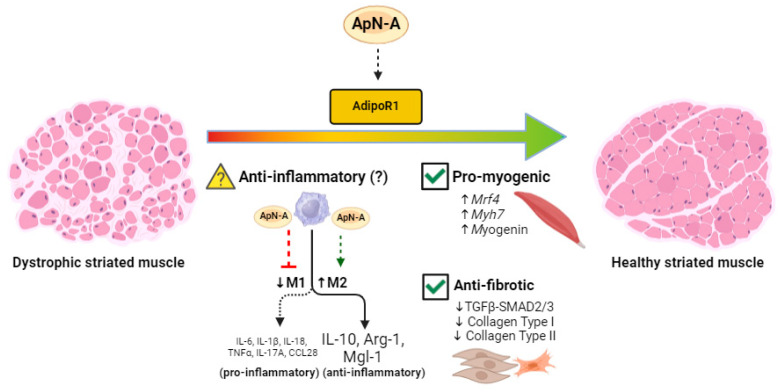
Proposed mechanisms linking ApN receptor agonism to improved inflammatory, myogenic, and fibrotic profiles of striated muscle in dystrophin-deficient mice. Dystrophic mice exhibit low circulating ApN levels [17] and restoring these levels with exogenous ApN agonists may shift the pro-inflammatory macrophage polarization [19] toward the anti-inflammatory polarization, characterized by elevations in IL-10, Arg-1, and Mgl-1, although this is yet to be proven in *mdx* mice (denoted by a question mark) [103,104,107,108]. Additionally, as previously demonstrated in *mdx* mice, ApN can improve the myogenic profile of the dystrophin-deficient skeletal muscle by upregulating *Mrf4*, *Myh7*, and *Myogenin* while simultaneously attenuating fibrosis by downregulating TGFβ-SMAD2/3, Collagen Type I, and Collagen Type II (check mark denotes *mdx* data) [21]. ApN-A = Adiponectin agonist. Made using www.biorender.com (accessed on 1 June 2024).

## Data Availability

No new data were created or analyzed in this study. Data sharing is not applicable to this article.

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
