# Peer review of "Recent Advances in Pre-Clinical Development of Adiponectin Receptor Agonist Therapies for Duchenne Muscular Dystrophy"

_biomedicines, 2024, doi:10.3390/biomedicines12071407_

Round 1

Reviewer 1 Report

Comments and Suggestions for Authors

This is a qualitative review from leaders in the field of studying the mechanisms of development of muscular dystrophies and, in particular, Duchenne dystrophy, as well as the development of promising approaches for the treatment of pathology. Indeed, given the complex nature of the pathology, along with genetic therapy (which currently has extremely limited use), it is necessary to create additional universal approaches. In this review, the authors focus on modulation of the adiponectin receptor, which shows anti-inflammatory, antifibrotic and promyogenic effects on tissue. The review is well structured and written, and is based on the literature of the last few years. I have a few comments and recommendations:

1. Regarding the figures. mdx mice are black in color. This is not the most significant problem, but it is better to correct the figures.

2. It is known that the pathology of DMD is accompanied by a significant disruption of ion homeostasis, this is due to dysfunction of ion channels of the sarcolemma, as well as dysfunction of the SR and mitochondria. This is not taken into account in the manuscript. It is important to note that adiponectin is able to activate calcium influx into muscle fibers. Dystrophin-deficient fibers are already quite overloaded with calcium. Will this be a problem?

3. If possible, I recommend covering in more detail the effect of adiponectin receptor modulation on cardiac pathology. Is there information about changes in the electrophysiological parameters of this organ when modulating this receptor?   

Author Response

Reviewer 1:

This is a qualitative review from leaders in the field of studying the mechanisms of development of muscular dystrophies and, in particular, Duchenne dystrophy, as well as the development of promising approaches for the treatment of pathology. Indeed, given the complex nature of the pathology, along with genetic therapy (which currently has extremely limited use), it is necessary to create additional universal approaches. In this review, the authors focus on modulation of the adiponectin receptor, which shows anti-inflammatory, antifibrotic and promyogenic effects on tissue. The review is well structured and written, and is based on the literature of the last few years. I have a few comments and recommendations:

  1. Regarding the figures. mdx mice are black in color. This is not the most significant problem, but it is better to correct the figures.

Response: Thank you for this comment. This aspect of the images (graphical abstract; Figure 2) has been remediated.

  1. It is known that the pathology of DMD is accompanied by a significant disruption of ion homeostasis, this is due to dysfunction of ion channels of the sarcolemma, as well as dysfunction of the SR and mitochondria. This is not taken into account in the manuscript. It is important to note that adiponectin is able to activate calcium influx into muscle fibers. Dystrophin-deficient fibers are already quite overloaded with calcium. Will this be a problem?

Response: We have added new sentences that discuss the role of ApN in influencing intramyocellular Ca2+ transients in both skeletal muscle and cardiomyocytes. Please see lines 246-267.

  1. If possible, I recommend covering in more detail the effect of adiponectin receptor modulation on cardiac pathology. Is there information about changes in the electrophysiological parameters of this organ when modulating this receptor?

Response: We are not aware of any available data/literature regarding ApN’s effects on cardiac electrophysiological parameters. This represents a gap in knowledge that requires further exploration. While the focus of this review is not to reiterate the well published literature on the role of ApN in modulating cardiac pathology, we have included several citations (See lines 243-245) to highlight seminal literature in this area. This new sentence has been added to refer readers to more extensive reviews expanding on our existing ApN-Cardiac pathology section.

Reviewer 2 Report

Comments and Suggestions for Authors

V

Manuscript ID: biomedicines-3014745

Recent advances in pre-clinical development of adiponectin receptor agonist therapies for Duchenne muscular dystrophy

Current knowledge about adiponectin as a therapeutic agent for DMD is scarce. It is also unclear by which pathophysiological mechanism adiponectin should work in dmd

Most data come from animal models.

Particularly memory fucntions were improved in theses models by stimulation of the adiponectin receptor.

It is suggested to generate more original data before reviewing inconclusive and preliminary data

05/24

Comments on the Quality of English Language

could be improvied

Author Response

Reviewer 2:

Comments and Suggestions for Authors

Manuscript ID: biomedicines-3014745

Recent advances in pre-clinical development of adiponectin receptor agonist therapies for Duchenne muscular dystrophy

  1. Current knowledge about adiponectin as a therapeutic agent for DMD is scarce. It is also unclear by which pathophysiological mechanism adiponectin should work in dmd

  1. Most data come from animal models.

Response: Thank you for these two comments. This is indeed a major purpose of the review with respect to stimulating discussion and promoting further research in this area.

  1. Particularly memory functions were improved in theses models by stimulation of the adiponectin receptor.

Response; We have added a new paragraph outlining recent findings from our group that ALY688 preserves recognition memory in D2.mdx mice. Please see lines 393-401.

  1. It is suggested to generate more original data before reviewing inconclusive and preliminary data

Response: We have added new clarification that the intent of this review is to stimulate discussion and activity in this new area of research. The review article format is relatively brief and appropriate for this intention. This edit is exemplified on lines 68-77.

Round 2

Reviewer 1 Report

Comments and Suggestions for Authors

I am satisfied with the author's responses to my questions/issues

Reviewer 2 Report

Comments and Suggestions for Authors

manusript has not significantl improvied compared to the previous verison 

Comments on the Quality of English Language

fair